# TranSPHIRE: automated and feedback-optimized on-the-fly processing for cryo-EM

Markus Stabrin [1], Fabian Schoenfeld[1], Thorsten Wagner [1], Sabrina Pospich [1], Christos Gatsogiannis [1] & Stefan Raunser [1✉]

Single particle cryo-EM requires full automation to allow high-throughput structure determination. Although software packages exist where parts of the cryo-EM pipeline are automated, a complete solution that offers reliable on-the-fly processing, resulting in high-resolution structures, does not exist. Here we present TranSPHIRE: A software package for fully-automated processing of cryo-EM datasets during data acquisition. TranSPHIRE transfers data from the microscope, automatically applies the common pre-processing steps, picks particles, performs 2D clustering, and 3D refinement parallel to image recording. Importantly, TranSPHIRE introduces a machine learning-based feedback loop to re-train its picking model to adapt to any given data set live during processing. This elegant approach enables TranSPHIRE to process data more effectively, producing high-quality particle stacks. TranSPHIRE collects and displays all metrics and microscope settings to allow users to quickly evaluate data during acquisition. TranSPHIRE can run on a single work station and also includes the automated processing of filaments.

---

[1] Department of Structural Biochemistry, Max Planck Institute of Molecular Physiology, Otto-Hahn-Straße 11, 44227 Dortmund, Germany. ✉email: stefan.raunser@mpi-dortmund.mpg.de

Single particle electron cryomicroscopy (cryo-EM) has successfully established itself as a prime method to determine the three-dimensional structure of macromolecular complexes at near-atomic resolution[1,2]. The technique has therefore the potential to become a key tool for drug discovery research[3]. However, single particle analysis (SPA) studies still require large amounts of processing time, expert knowledge, and computational resources. With the number of modern high-throughput microscopes growing rapidly, there is an urgent demand for a robust, automated processing pipeline that requires little to no user intervention. This need is felt especially in the field of drug discovery[3]. Automated data processing does not necessarily aim at the highest possible resolution. However, resolutions at 3–4 Å already allow identifying the position of small molecules in the binding pocket of a protein, especially when ligand-free high-resolution structures are available as exemplified by previous data of the TRPC4 channel and actin filaments[4,5]. Consequently, automated processing especially facilitates the effective screening of many drug candidates. Recent advances in both hardware and software, which led to the first 1.2 Å resolution structures[6,7], further demonstrate the potential of cryo-EM for drug discovery research in the near future.

In many cases, data sets that were recorded for several days and can include 10,000–20,000 movies turn out to be unusable for high-resolution structure determination during subsequent data processing. It is therefore necessary for users to obtain feedback on the quality of their data immediately during recording. This enables them to decide whether or not to continue a session, adjust any of the acquisition parameters at the microscope, and compare different grids. This can only be achieved when processing the data in parallel to data acquisition. A fully-automated pipeline requires streamlined data transfer, automated pre-processing and processing workflows, free of any user bias.

Several software packages partially address these issues. For example, CryoFLARE[8], Focus[9], Scipion[10,11], Appion[11], WARP[12], RELION IT[13], and CryoSPARC Live perform live analysis and processing in parallel to data acquisition. This is done by chaining together the different tools of the single particle analysis pipeline, but it lacks any automatic data optimization. The Cianfrocco lab recently published a deep learning-based pre-processing routine that can filter particles of insufficient quality automatically. Their approach is based on 2D classification results analyzed by a deep learning based classifier[14]. However, this tool has not been designed to run live during data acquisition. The Liu lab went one step further and published a self-supervised workflow for particle picking[15]. Their software automatically selects 2D class averages based on the %/Res value provided by Relion[16] and uses them to re-train their model for particle picking. Although significant progress has been made towards both automatic data optimization and on-the-fly processing, a fully automated SPA pipeline that facilitates data analysis and data optimization in parallel to data acquisition is still missing.

Here we present TranSPHIRE, a fully automated pipeline for on-the-fly processing of cryo-EM data. It combines deep learning tools for particle picking and 2D class selection with a novel, feedback-driven approach to re-train the integrated crYOLO particle picker[17] during ongoing pre-processing. This allows TranSPHIRE to perform GPU accelerated 2D classification to provide high-quality 2D class averages and, subsequently, 3D reconstructions from clean data. This gives experimentalists the means to quickly evaluate both the quality of their data sets as well as their chosen microscope settings during data acquisition. A combination of new and improved tools allows TranSPHIRE to provide users with the strongest early results in the shortest amount of time, without the need for user intervention. While TranSPHIRE can run on a single GPU machine, it additionally offers the possibility to outsource the computationally expensive 3D reconstructions via SSH connection to a separate machine or computer cluster. Importantly, it allows users to perform automated high-throughput on-the-fly screenings for different buffer conditions or ligands of interest as well as to fine-tune the workflow for the respective target-protein and perform digital purification during image acquisition.

## Results

**General setup, functionality, and layout of TranSPHIRE.** TranSPHIRE is an automated pipeline for processing cryo-EM data sets (Fig. 1). It is developed in Python 3 to run on Linux, and is available online for free. TranSPHIRE performs parallelized data transfer and flexibly integrates a range of commonly used pre-processing tools, as well as the advanced processing tools of the SPHIRE package[18]. Using these tools, TranSPHIRE implements a fully-automated pipeline to process cryo-EM data on-the-fly during data acquisition. TranSPHIRE is designed to allow users to make the best use of their available resources by prioritizing data analysis, presenting early results, and using machine learning tools to identify and process only those parts of the data that contribute to high-quality results.

TranSPHIRE is controlled via an easy-to-use GUI that allows users to set up a session, and choose and configure the desired tools to use (Supplementary Fig. 1 and Table 1). For pre-processing, the TranSPHIRE pipeline integrates MotionCor2[19] and Unblur[20] for beam induced motion correction with dose weighting; as well as CTFFIND4[21], CTER[22], and GCTF[23] for CTF estimation. This modularized integration is entirely parameterized, allowing experimentalists to both choose their preferred tools as well as configure them as needed—all without leaving the TranSPHIRE GUI. Available parameters are sorted by level of usage ("main", "advanced", and "rare") to highlight and help identify the most commonly adjusted parameters for each tool.

During the session, TranSPHIRE automatically parallelizes the batch-wise processing of incoming micrographs, outsources computationally expensive steps to available GPUs, and produces preliminary 2D class averages and 3D reconstructions based on the most recently processed batch of data (Fig. 2 and Supplementary Fig. 2). Specifically, every process type, e.g. motion correction, CTF estimation, and particle picking, runs in parallel to each other in separate threads. Every thread monitors its respective queue and starts processing of new data upon its arrival. Afterwards, the output data is put into the queue of the consecutive processing step. To allow for efficient usage of the available resources, an optimized queueing system has been developed within TranSPHIRE to distribute the jobs which run on GPUs (Table 1). This avoids oversubscription of the limited GPU memory. In contrast, oversubscription of available CPU resources does usually not lead to a loss in performance, therefore the task of CPU resource scheduling is left to the operating system. In addition to the actual computation, each thread performs organizational tasks like creating the command of the process or to check the output for errors and values that are outside of the specified range. Since these tasks are sometimes time consuming but not computational intensive, the number of assigned threads for all tasks within TranSPHIRE usually exceeds the maximum available CPUs of the respective workstation to run at maximum speed (also see Fig. 2).

Through this optimal distribution of processes, TranSPHIRE runs on-the-fly for a wide range of data acquisition settings using a single workstation (Supplementary Fig. 3; see "Methods" for details about hardware). Moreover, TranSPHIRE can catch-up with the speed of the acquisition after the initial delay due to the feedback loop (see below) for routinely used data acquisition

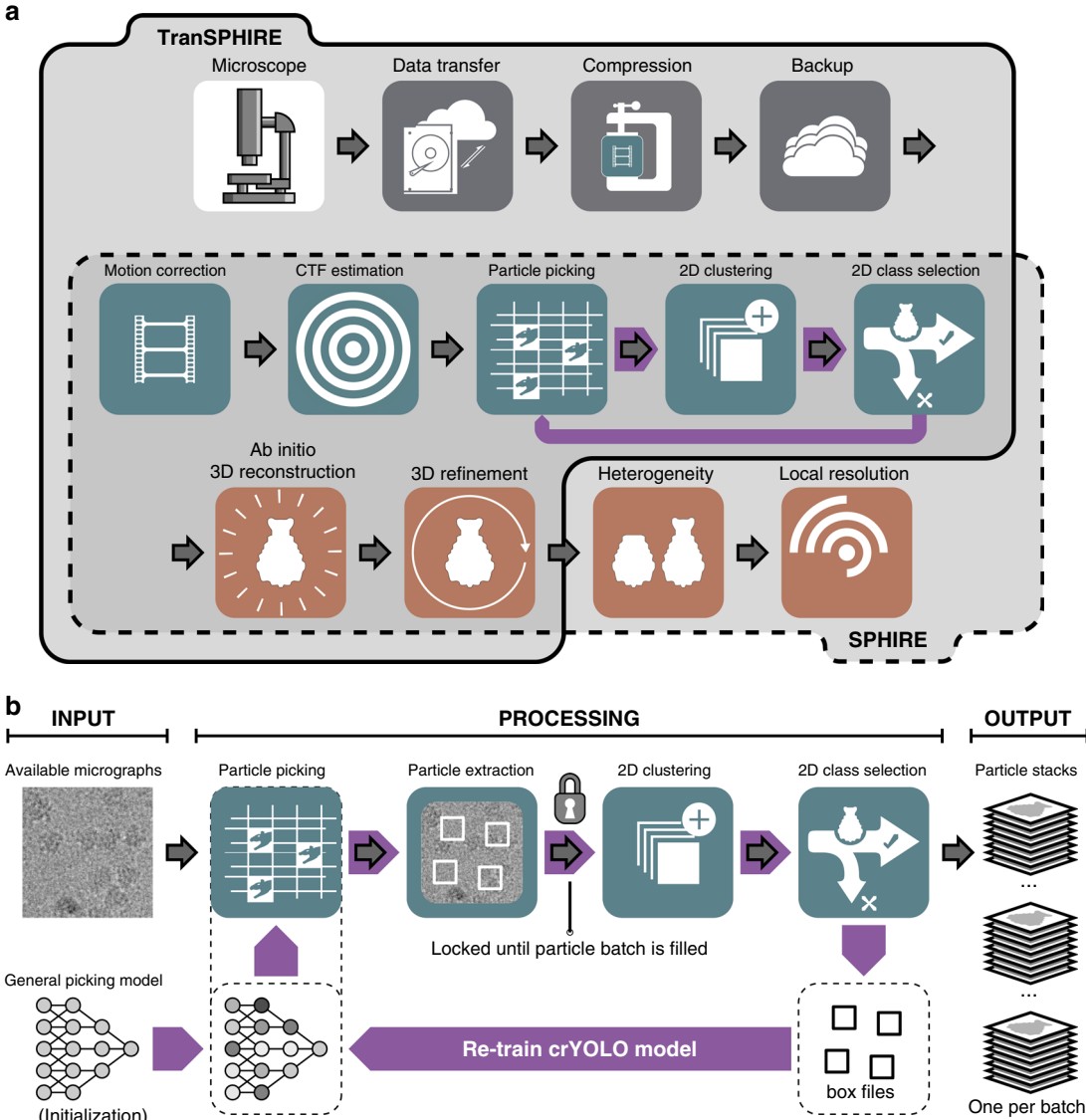

**Fig. 1 The TranSPHIRE pipeline and the SPHIRE backend. a** Upper register (solid line): Overview of the integrated TranSPHIRE pipeline and all automated processing steps. The pipeline includes file management tasks, i.e., parallelized data transfer, file compression, and file backup (gray); 2D processing, i.e., motion correction, CTF estimation, particle picking, 2D clustering, and 2D class selection (turquoise); and 3D processing, i.e., ab initio 3D reconstruction and 3D refinement (red). Additionally, the pipeline includes an automated feedback loop optimization to adapt picking to the current data set during runtime (purple). Lower register (dotted line): The SPHIRE software package forms the backend for TranSPHIRE and offers the tools used for 2D and 3D processing. SPHIRE includes additional tools for advanced processing, such as heterogeneity analysis and local resolution determination. **b** The TranSPHIRE feedback loop. Gray arrows indicate the flow of data processing. Purple arrows indicate the flow of the feedback loop. Left (input): Micrographs are initially picked using the crYOLO general model. Center (processing): Particles are picked and extracted. Once a pre-defined number of particles have been accumulated, the pipeline performs 2D classification; the resulting 2D class averages are labeled as either "good" or "bad" by Cinderella. Class labels and crYOLO box files are then used to re-train crYOLO and adapt its internal model to the processed data. In the next feedback round this updated model is used to re-pick the data. Right (output): After five feedback rounds, the complete data set is picked with the final optimized picking model and 2D classified in batches. For every batch a particles stack of "good" particles is created and available for 3D processing.

schemes (Supplementary Fig. 3c). Thus, initial 2D class averages and 3D reconstructions are available within a few hours after starting the data collection (Fig. 2 and Supplementary Fig. 3).

Throughout the processing, TranSPHIRE collects all data quality metrics produced by its individual tools, links them with the relevant micrographs where appropriate, and presents them front and center in its GUI (Supplementary Fig. 1).

Optionally, notifications for early milestones such as 2D class averages and preliminary 3D maps, can also be sent via email. These features enable experimentalists to both identify and address any issues as soon as they surface during data acquisition, without requiring constant user supervision. Additionally, all results produced by the integrated tools during processing are also copied in parallel to the pre-defined workstation and backup locations (Supplementary Figure 1). To support interoperability with existing packages, all pre-processing steps until particle picking support the file formats used in both SPHIRE and RELION; for later processing steps the SPHIRE package[18] provides utilities to easily convert SPHIRE files into RELION[16] .star files, which can then be used for further processing in many other cryo-EM software tools.

**Transfer and pre-processing.** Once a session starts, TranSPHIRE automatically detects and transfers new micrographs from the camera computer of the microscope (Fig. 1 and Supplementary Fig. 2). These data are moved in parallel to several, user-specified locations e.g. a work station or cluster for processing, and a backup storage server. In case of the latter, TranSPHIRE also automatically compresses the data to preserve storage space. Copy locations may also include additional spaces such as transportable hard discs. If desired, TranSPHIRE further renames files, and deletes images from the camera workstation in order to free up more space to enable continuous data collection. It also extracts meta data, such as acquisition time, grid square, hole number and coordinates, spot scan, and phase plate position from .xml files provided by EPU (FEI Thermo Fisher Scientific) or .gtg files provided by Latitude S (Gatan) (for details see Table 2).

During the ongoing data transfer, any data that has already been copied is pre-processed in parallel (Fig. 2 and Supplementary Fig. 2). During setup, users can choose to perform motion correction using either MotionCor2[19] or Unblur[20]. While motion correction is performed, TranSPHIRE presents all relevant metrics, such as the average shift per frame, or the overall shift per micrograph (Supplementary Fig. 1). For CTF estimation, users can set up TranSPHIRE to use either CTFFIND4[21], CTER[22], and GCTF[23]. Depending on whether or not CTF estimation on movies is activated in TranSPHIRE, CTF estimation is performed in parallel to motion correction or afterwards (Supplementary Fig. 2). The metrics extracted and displayed by TranSPHIRE include defocus, astigmatism, and the resolution limit (Supplementary Fig. 1). Combined with the information gathered during motion correction, these values allow experimentalists to assess the performance and alignment of the microscope during acquisition, and adjust any thresholds to automatically discard low-quality micrographs as necessary.

For particle picking the TranSPHIRE pipeline integrates crYOLO[17], our state of the art deep learning particle picker. During picking, TranSPHIRE displays the particles picked per micrograph, which allows users to assess the picking performance and overall sample quality (Supplementary Fig. 1).

Once a fixed threshold of picked particles is reached (Supplementary Fig. 4; also see "Methods"), TranSPHIRE launches 2D classification using a GPU accelerated version of ISAC2[24] (Fig. 1). ISAC2 limits the number of class members to spread the given particles across multiple classes which prevents individual classes from growing too large. This results in sharp, equal-sized, and reproducible classes that contain all possible orientations exceeding the minimum class size. They enable experimentalists to reliably assess particle orientations and overall quality, and help to identify possible issues such as preferred orientations or heterogeneity.

**Table 1 Hardware utilization for software tools supported by TranSPHIRE.**

| Task | Program name | GPU | CPU | Reference |
|---|---|---|---|---|
| Motion correction | Unblur | | X | 20 |
| | MotionCor2 | X | | 19 |
| CTF estimation | CTER | | X | 22 |
| | GCTF | X | | 23 |
| | CTFFIND4 | | X | 21 |
| Particle picking | crYOLO | X | | 17,34 |
| 2D classification | GPU ISAC | X | | Based on ISAC[24] |
| 2D class selection | Cinderella | X | | 25 |
| 3D ab initio reconstruction | RVIPER | | X | 18 |
| 3D refinement | MERIDIEN | | X | 18 |
| Utilities programs | SPHIRE | | X | 18 |
| | EMAN2 | | X | 27 |
| | IMOD | | X | 35 |

List of all currently supported software tools indicating if the given software runs on CPUs or GPUs within the TranSPHIRE pipeline. Tools are grouped corresponding to the pipeline task they are utilized for.

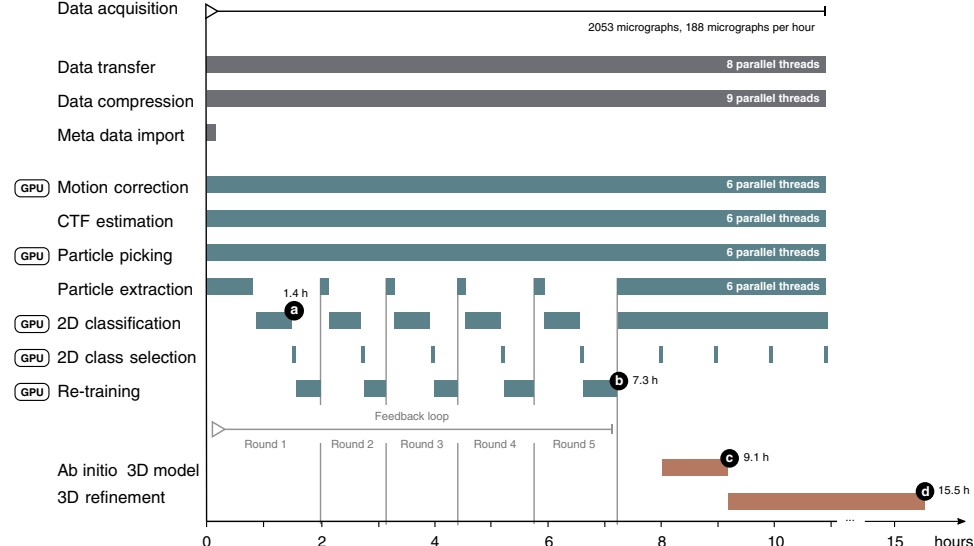

**Fig. 2 Timeline of the TranSPHIRE pipeline.** Timeline depicting the parallel execution of the processes of the TranSPHIRE pipeline. Timings are based on a Tc holotoxin data set consisting of 2053 micrographs, each containing 36 particles on average, collected at a speed of 188 micrographs per hour (K2 super-resolution, 40 frames). TranSPHIRE ran on-the-fly up to the creation of an ab initio 3D reconstruction using default settings. Important milestones are denoted in black: **a** first 2D class averages produced after 1.4 h; **b** end of the feedback loop after 7.3 h; **c** ab initio 3D reconstruction after 9.1 h; and **d** final 3D reconstruction of the first batch of particles after 15.5 h. Due to the internal scheduling of modern operating systems, and because not every TranSPHIRE thread is always working to capacity, the number of available CPUs (12/24 hyperthreading) and assigned TranSPHIRE threads (45) is not identical, and does not limit the speed of the computations.

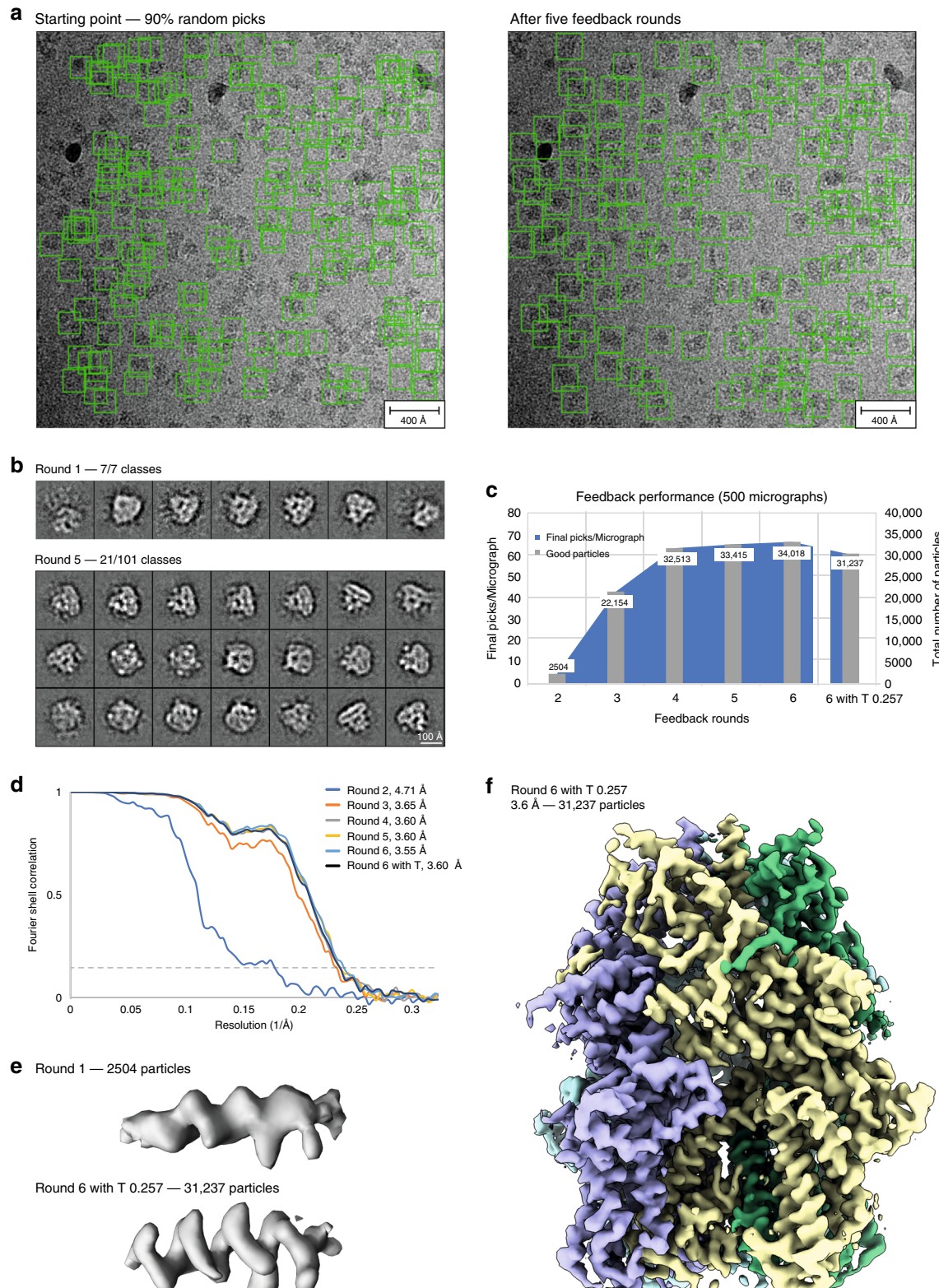

**Fig. 3 Processing the TRPC4 membrane channel using a deliberately hampered picking model. a** To simulate low quality picking, only 10% of the initial crYOLO picks were used while the remaining 90% were re-positioned randomly (left). After the feedback loop crYOLO reliably picks the TRPC4 particles (right). **b** Total amount of 2D class averages produced in the first iteration of the feedback loop (top) and 21 representative averages produced in the final iteration of the feedback loop (bottom). **c** Progression of the number of particles labeled "good" when applying the intermediate picking models of the feedback loop to a fixed subset of 500 micrographs. The curve flattens out in the final iterations, indicating the convergence of the feedback loop optimization. **d** Fourier shell correlation (FSC) curves of the individual 3D reconstructions computed from particles labeled "good" (also see **c**). **e** Representative α-helix (amino acids 600–615) illustrating the improvement of the density when using the final (bottom) compared to the initial (top) picking model. **f** 3D reconstruction of TRPC4 computed from 500 micrographs using the optimized picking model.

**Table 2 Overview of all meta information extracted by TranSPHIRE.**

| Meta data | EPU | Latitude S |
|---|---|---|
| Stage coordinates | X | X |
| Microscope defocus | X | X |
| Applied defocus | X | X |
| Pixel size | X | X |
| Exposure time | X | X |
| Number of fractions | X | X |
| Image dimensions | X | X |
| Microscope voltage | X | |
| Microscope dose | X | |
| Phase plate information | X | |
| Super resolution mode | X | |
| Gridsquare number | X | X |
| Hole ID | X | X |
| Spotscan ID | X | X |
| Acquisition time | X | X |
| Beam tilt information | X | |

List of meta data automatically extracted by TranSPHIRE during data acquisition using either EPU (FEI Thermo Fisher Scientific) or Latitude S (Gatan).

The 2D class averages are then sorted by Cinderella[25], our integrated deep learning tool for 2D class selection. Cinderella labels the given 2D classes as either "good" or "bad" and determines which class averages and, thereby, particles are used for further processing. This results in an automatic cleaning of the data and allows TranSPHIRE to process only the relevant subset of a given data set, thereby dramatically lowering the amount of data processed by the computationally expensive steps of 3D reconstruction and refinement. Since a working Cinderella model is essential for the success of the following feedback loop, it is important to correct an inadequate Cinderella model as quickly as possible. For this purpose, a built-in tool to re-train Cinderella is available within the TranSPHIRE GUI which enables the user to iteratively adjust the model to the specific needs of the data at hand. Specifically, all 2D class averages that have been generated up to this point can be manually labeled as either "good" or "bad" and then used for re-training Cinderella. A reliable model can usually be produced from 40 manually labeled classes, that should be approximately evenly distributed into "good" and "bad". As soon as the re-training is complete, TranSPHIRE shows a preview of the 2D selection using the new Cinderella model. The model can then be further refined if required or green-lit to be used in the TranSPHIRE pipeline. For data sets that require re-training of Cinderella, it is usually sufficient to re-train once on the very first set of obtained 2D class averages. However, further re-training can be performed if deemed necessary by the user to adapt to the 2D class averages of higher quality produced in later stages of the feedback loop. In case of sample heterogeneity, which cannot be accounted for on a 2D level, it is important to run a 3D sorting procedure after data acquisition. An FAQ section for common pitfalls and problems is available in the TranSPHIRE wiki (https://transphire.readthedocs.io).

**Feedback loop to optimize particle picking**. For any cryo-EM pipeline the ability to reliably perform high-quality picking irrespective of the data at hand is essential. This poses a challenge when processing is to be automated, as this immediately excludes any user intervention such as manual inspection of the picking results. The latter is especially relevant if a sample is unknown to the picking procedure, or is otherwise difficult to process, e.g. due to contamination or interfering conformational states – issues that usually need to be identified by a qualified expert before processing can continue.

TranSPHIRE solves these issues by introducing a machine learning-based feedback loop that repeatedly re-trains the fully-integrated crYOLO[17] deep learning particle picker during data acquisition to adapt picking to the given data set (Fig. 1b). This enables crYOLO to specifically target those particles that end up in stable 2D class averages, while, at the same time, learning to disregard particles that do not. First, incoming motion corrected micrographs are forwarded to crYOLO for picking. Once a batch of 20,000 picked particles has been accumulated, it is handed over to our GPU accelerated version of the 2D classification algorithm ISAC2[24] (Supplementary Fig. 3). Here we determine which particles can be used to create stable 2D class averages, and reject the particles that cannot be accounted for. The newly produced 2D class averages are given to our deep learning tool Cinderella[25], which labels each class average as either "good" or "bad". At this point, the particles of the "good" classes are used to re-train crYOLO and update its internal model. Specifically, we randomly select a maximum of 50 micrographs that contain particles that ended up in the "good" classes for the re-training (for details see "Methods"). Once the training and thus the first feedback round has completed, processing re-starts using the optimized picking model (Fig. 1b).

The TranSPHIRE feedback loop iterates five times, which has proven sufficient to achieve convergence in our experiments. As a consequence, this is not repeated for the remainder of the data acquisition. Re-training crYOLO[17] to become increasingly more proficient at targeting particles that end up in "good" classes has the additional benefit of trimming down the overall size of the data set. Though the pre-processing of cryo-EM data is already time consuming, the following 3D refinement requires even more computational power. While it is usually customary to process as much data as possible, the computational cost of 3D refinement usually does not scale linearly, and such an approach will not be sustainable in the near future. This limitation in scalability is further exacerbated by the fact that image acquisition speeds and sizes of data sets are both growing rapidly. Therefore, the aim should be to process as little and as homogeneous data as possible, without harming the quality of the final reconstruction. Fortunately, it is known that cryo-EM data sets contain a large amount of unusable data that can be safely discarded—if we have a way to reliably ensure that we keep those data that we are actually interested in. The TranSPHIRE feedback loop offers this functionality and provides quality in quantity.

**Ab initio 3D model reconstruction and 3D refinement**. To compute a 3D reconstruction, the particles included in all classes labeled "good" by Cinderella are extracted and form a clean, high-resolution particle stack. If there is no initial 3D reference provided to TranSPHIRE, the pipeline waits until at least 200 (by default) "good" classes have been accumulated. The respective 2D class averages are then used to create a reproducible, ab initio 3D reconstruction using SPHIRE RVIPER[18,26] (Fig. 1). This provides a first view of the structure of the target protein and a first impression of the conformational state. In case an external starting model, either map or atomic model, is already available, the SPHIRE and EMAN2 package[18,27] provides straight-forward tools to convert, rescale and clip it to the desired box and pixel size.

The initial 3D reference is then used by TranSPHIRE to initialize the 3D refinement using SPHIRE MERIDIEN (Fig. 2). While the initial map is computed only once, a new 3D refinement is started every time another set of 40,000 (by default) "good" particles has been accumulated.

**Table 3 Summary of the feedback loop statistics for the TRPC4 data.**

| Feedback round | Good classes | Good particles | Picks/Mic | Good picks/Mic | Resolution | Relative good picks |
|---|---|---|---|---|---|---|
| 2 | 28 | 2504 | 20 | 5 | 4.71 | **0.25** |
| 3 | 236 | 22,154 | 104 | 44 | 3.65 | **0.43** |
| 4 | 349 | 32,513 | 132 | 65 | 3.60 | **0.49** |
| 5 | 355 | 33,415 | 152 | 67 | 3.60 | **0.44** |
| 6 | 361 | 34,018 | 147 | 68 | 3.55 | **0.46** |
| 6 + T 0.257 | 331 | 31,237 | 114 | 62 | 3.6 | **0.55** |

For every feedback round as well as the final run after optimization of the picking threshold (6 + T x.xx) the number of classes labeled "good" by Cinderella; the number of particles included in these classes; the total number and the number of good particles picked per micrograph; the final resolution of the 3D reconstruction; and the relative amount of good particles (highlighted in bold) are listed for 500 micrographs.

Note that in contrast to SPHIRE RVIPER, which only uses the first 2D class averages, SPHIRE MERIDIEN uses all particles subsumed by the last batch of "good" particles. The fully-automated creation of an initial 3D map and continuous production of a series of refined reconstructions based on that latest data enables TranSPHIRE to present high-resolution structures already during data acquisition.

This enables for a more detailed, on-the-fly evaluation by the user, such as analyzing the conformational state and/or confirming whether and where a ligand is bound if a ligand-free high-resolution reference structure is present. By providing a series of reconstructions—one for every batch, TranSPHIRE also offers a time-resolution of the data set, enabling experimentalists to gauge the quality of their data over time throughout data acquisition.

With the following three experiments we illustrate the capabilities of TranSPHIRE to automatically adapt to unknown data, make use of prior knowledge to selectively target the conformational subpopulation within a sample and process filamentous data.

**Learning to pick a membrane channel without user intervention**. Similar to crYOLO, many modern particle picking programs are based on machine learning, where an internal model is trained to recognize particles within micrographs[12,28,29]. While this method features an inherent capacity to generalize to unseen data sets, this ability is limited. Therefore, reliable picking can usually not be guaranteed out of the box when samples differ too much from the original training data of the network. Samples might also be of unusually low contrast, or an unknown form of contamination is encountered. While such issues can be overcome by adding the problematic data to the training set, this requires manual user intervention on multiple levels. First, the insufficient picking capability has to be detected; second, an experienced experimentalist has to pick a small amount of training data by hand; and third, the network has to be re-trained manually.

The TranSPHIRE feedback loop resolves this issue and entirely foregoes the need for user intervention even when facing data that is either unknown to the picking model or yields insufficient picking results for any other reason. To demonstrate this ability, we processed a data set of the TRPC4 membrane protein channel with the TranSPHIRE feedback loop using a picking model without any prior knowledge of this protein (Fig. 3). Specifically, to ensure the sample was unknown to crYOLO at the start of the feedback loop, we removed all four TRP channel data sets normally included in the training data of the crYOLO general model. Additionally, in order to simulate a bad generalization of crYOLO we randomized 90% of all picks in the first iteration of the feedback loop (Fig. 3a). This was done by replacing 90% of the particle boxes determined by crYOLO with randomly positioned boxes within the same micrographs. In combination, these measures ensured that the initial picking results were almost

entirely unusable and successful re-training had to take place in order to enable further processing of the data.

Despite the bad starting point, by the final feedback loop iteration the repeatedly re-trained model has successfully learned to pick the previously unknown TRPC4 particles resulting in high-resolution 2D class averages (Fig. 3a, b). An evaluation of the performance of the feedback loop on a fixed subset of 500 micrographs (see Methods for details), illustrates that the number of "good" particles increases sharply within the early iterations of the feedback loop from an initial 25% of particles to a stable value of ~50%, (Table 3 and Fig. 3c) and a final resolution of 3.6 Å (FSC = 0.143). This increased ability to identify a greater number of usable particles on the same subset of micrographs is also reflected in the map quality and achieved resolution when using the intermediate crYOLO models produced during the individual feedback rounds to process the fixed set of 500 micrographs (Fig. 3d–f).

This experiment furthermore demonstrates the ability of crYOLO to adapt to unknown data even if only sparse training data is available. In the initial round of the feedback loop a mere 5 particles per micrograph ended up in "good" classes on average – and, consequently, are all that was available to re-train the picking model (Table 3).

In summary, the TranSPHIRE feedback loop is able to automatically optimize the internally used picking model and provide reliable, high quality picking results even when processing challenging samples that initially are barely recognized by the model. We have shown that in such a case, after five feedback rounds, crYOLO is able to pick the TRPC4 membrane protein to completion, without requiring the user to continuously monitor, let alone disrupt the ongoing data processing. The feedback loop optimization is fully-integrated into the TranSPHIRE pipeline and works entirely automated out of the box. Its capabilities extend to difficult data sets such as membrane proteins, and enable advanced processing methods, such as targeting specific conformational states, or processing filamentous data sets, as demonstrated in the following.

**Targeting a conformational state using prior knowledge**. A basic assumption of most algorithms currently used to process cryo-EM data is that all particles in a data set are projections of the same structure, hidden behind a curtain of noise. In reality, however, cryo-EM samples are often more complex, and can contain multiple conformational states of the target structure, impurities, and aggregates. Filtering such unwanted data and selectively targeting only a subset of the structures found within a sample is one of the fundamental issues in cryo-EM, and often requires significant efforts to address and resolve.

The TranSPHIRE feedback loop offers a straightforward solution to this issue by allowing the injection of additional knowledge into the pipeline, either before or during runtime. This enables users to incorporate and make use of information that is

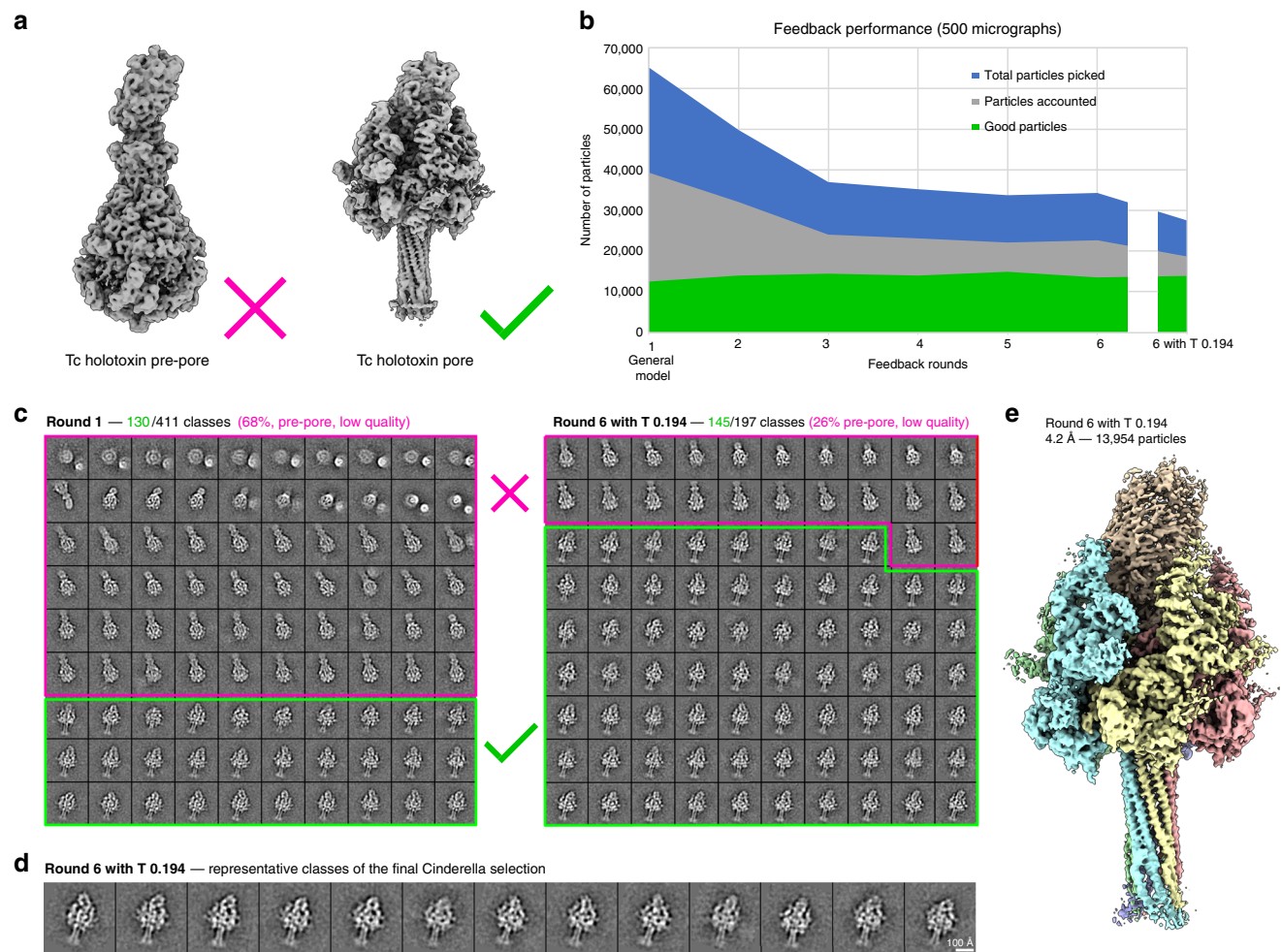

**Fig. 4 Using prior knowledge to extract a pre-selected conformational state. a** The processed data set contains the Tc holotoxin in both the pre-pore state (left) and the more rare pore state (right). In this experiment, we specifically target the pore state. **b** Progression of the number of picked particles (blue), those accounted during 2D classification (gray) and particles labeled "good" i.e. representing the pore state (green) when applying the intermediate picking models of the feedback loop to a fixed subset of 500 micrographs. Initial picking is dominated by pre-pore state particles. This overhead is reduced with each iteration, while the amount of picked pore state particle remains stable. **c** Representative 2D class averages depicting the decrease of unwanted classes (pore state or low quality; marked magenta) from an initial 68% in the first feedback round (left) to 26% after the last feedback round (right). **d** Representative 2D class averages depicting the pore state as selected by Cinderella in the final iteration of the feedback loop. **e** 3D reconstruction of the Tc holotoxin pore state computed from 500 micrographs using the final optimized picking model.

already available, as well as information that was just produced during acquisition. Specifically, a set of 2D class averages of the target structure can be used to train Cinderella[25] to only recognize these averages as representatives of "good" classes, and, consequently, everything else as "bad". If such averages are available beforehand, Cinderella can be pre-trained; otherwise, the feedback loop can be paused once the first set of 2D class averages are produced in the TranSPHIRE pipeline and continued after manual re-training of Cinderella. This additional training step to embed additional knowledge into the TranSPHIRE pipeline enables us to steer the re-training of the picking model during the feedback loop iterations. More precisely, particles that end up in sharp classes depicting a different particle, a subcomplex, and/or the target protein in the wrong conformational state (for example) will now also be labeled as "bad" by Cinderella, despite their high quality. During the feedback loop, crYOLO will thus be taught to only focus on particles that end up in quality classes depicting the wanted particle or state, while, at the same time, reject anything else, including sharp classes from an unwanted subpopulation.

To demonstrate the capability of the TranSPHIRE feedback loop to use prior knowledge and target a pre-selected conformation, we processed a sample of the Tc holotoxin that contained particles in two conformational states, namely the pre-pore and pore state (Fig. 4a). Of these, we only targeted the pore state, which is significantly more difficult to find as it only accounts for ~19% of the particles within the data set (Table 4 and Fig. 4b). Cinderella was trained with 318 examples of "good" classes (side-views of the pore state) and 664 examples of "bad" classes (views of the pre-pore state and contamination). During the feedback loop crYOLO was then re-trained with only those particle picks that ended up in "good" classes showing views of the pore state.

To evaluate the performance of the feedback loop we used the intermediate picking models produced during the individual feedback rounds to separately process a fixed set of 500 micrographs once the feedback loop had finished (see Methods for details). We observed a steady decrease of particles representing the pre-pore state—that we are not interested in— together with an initial rise and then level amount of pore state picks (Fig. 4b and Table 4). While initially only 19% of the

**Table 4 Summary of the feedback loop statistics for the Tc holotoxin data.**

| Feedback round | Good classes | Good particles | Picks/Mic | Good picks/Mic | Resolution | Relative good picks |
|---|---|---|---|---|---|---|
| 1 | 130 | 12,595 | 130 | 25 | 4.28 | **0.19** |
| 2 | 145 | 10,406 | 100 | 21 | 4.24 | **0.21** |
| 3 | 151 | 14,534 | 74 | 29 | 4.36 | **0.40** |
| 4 | 146 | 14,081 | 71 | 28 | 4.28 | **0.40** |
| 5 | 155 | 14,935 | 68 | 27 | 4.28 | **0.40** |
| 6 | 140 | 13,566 | 69 | 27 | 4.24 | **0.39** |
| 6 + T 0.194 | 145 | 13,954 | 55 | 28 | 4.24 | **0.50** |

For every feedback round as well as the final run after optimization of the picking threshold (6 + T x.xx) the number of classes labeled "good" by Cinderella; the number of particles included in these classes; the total number and the number of good particles picked per micrograph; the final resolution of the 3D reconstruction; and the relative amount of good particles (highlighted in bold) are listed for 500 micrographs.

particles resembled the pore state, slightly more than 50% of all picks ended up in 2D class averages depicting our targeted conformation when using the final optimized picking model (Fig. 4c, d). As in the previous experiment, the percentage of relative good picks per micrograph steadily increases. Notably, this happens while neither the number of good classes, nor the number of good particles seem to follow suit (Table 4). This means that our re-training efforts are working as intended: Over the course of the feedback loop, crYOLO learns to discard quality class averages of the pre-pore state that we are not interested in and instead focus on picking the less common pore state. Consequently, the amount of picked particles changes slowly, while, at the same time, the relative amount of "good" particle picks steadily increases, resulting in a 4.2 Å (FSC = 0.143) 3D reconstruction of the pore state from no more than 500 micrographs (Fig. 4e).

Taken together, these results illustrate how additional knowledge can be used to pre-train Cinderella, allowing TranSPHIRE to steer the re-training of the picking model during the feedback loop and to target a known subpopulation within the data. Using a picking model optimized for a specific conformation offers a two-fold advantage. First, reconstruction efforts will be more effective, as we gain more particles of the subpopulation that we are interested in. Second, reconstruction efforts will be more efficient, as the rejection of particles that end up in "bad" classes significantly shrinks the overall size of the data set. In our example, we reduce the number of picked particles from an initial total of 67,117 to a set of only 27,646 particles, without reducing the achieved resolution or the number of pore state particles that we are interested in (Fig. 4b). Any follow-up computations, such as costly 3D reconstructions, benefit greatly from such a reduction in data set size as it results in much more efficient use of the available computational resources.

**Automatic processing of filamentous proteins**. Filamentous proteins such as the actomyosin complex are notoriously difficult to process. This is because their structure is by definition not limited to a single element but rather forms a continuous strand that both enters and exits the enclosing frame of any picked particle image. Consequently, filamentous proteins are traced, rather than picked, and overlapping segments have to be identified along each filament, while filament crossings and contamination need to be avoided. In addition, filamentous projections share a similar overall geometry which increases the correlation between any two particles and interferes with alignment attempts during 2D classification. While there are several programs available that implement manual filament processing[18,30–33], until now there has not yet been any cryo-EM software package that offers the automated processing of filamentous data sets.

With TranSPHIRE we introduce a comprehensive software package for cryo-EM that includes the ability to automatically process filamentous proteins utilizing methods of the SPHIRE package[18]. While the actual processing is fully-automated, some preparation is still needed when using the TranSPHIRE pipeline to process filaments. Specifically, crYOLO needs to be trained to pick filaments[34], as these look fundamentally different from the single particle complexes known to its default general model. Additionally, Cinderella[25] also needs to be trained with 2D class averages of the filament in question. If such class averages are not available initially, the feedback loop can be halted for re-training Cinderella as soon as TranSPHIRE has produced them. Once the models for the deep learning decision makers of the pipeline are trained on the specific filamentous data, TranSPHIRE and its integrated feedback loop are ready to automatically process the respective filamentous data sets.

As an example of processing initially unknown filamentous data, we chose an actomyosin complex. To further demonstrate the ability of the feedback loop to adjust the picking to a specific filamentous protein complex, we trained crYOLO with multiple data sets of F-actin, which looks substantially different than the actomyosin complex (Fig. 5a). Thereby, crYOLO learns to trace filaments, but does not readily recognize actomyosin filaments resulting in a weak initial picking performance (Fig. 5b, c).

As soon as the first 2D class averages became available, the feedback loop was halted and a new Cinderella model was trained manually. Afterward the feedback loop continued through its default five iterations, automatically teaching crYOLO to identify projections of the actomyosin complex. To evaluate the performance of the feedback loop we separately processed a fixed set of 100 micrographs using the intermediate picking models produced during the individual feedback iterations (Fig. 5 and Table 5, see "Methods" for details).

Initially, a low confidence threshold of 0.1 (default) was used for picking in order to gather enough training data (Fig. 5c, d). However, the amount of picked particles and the confidence in the picks increased throughout the feedback loop (Fig. 5b, c). Thus, the picking threshold was adjusted to the default value of 0.3 after the feedback in order to exclude low confidence picks of contamination and filament crossings (Fig. 5b, d, e). Thereby, the number of relative good particles could be increased from 50% to 76% (Table 5) resulting in few classes labeled "bad" (Fig. 5f). The improvement is also visible when comparing the initial and final 3D reconstruction computed from the same set of 100 micrographs (Fig. 5g, h). Particularly, the final reconstruction of 4.4 Å (FSC = 0.143) is sufficient to identify a small molecule bound to the filament, highlighting how TranSPHIRE can simplify ligand screenings.

Using the feedback loop, TranSPHIRE offers the first cryo-EM software package that is able to automatically process filamentous data, even if the precise shape of a specific filament is initially

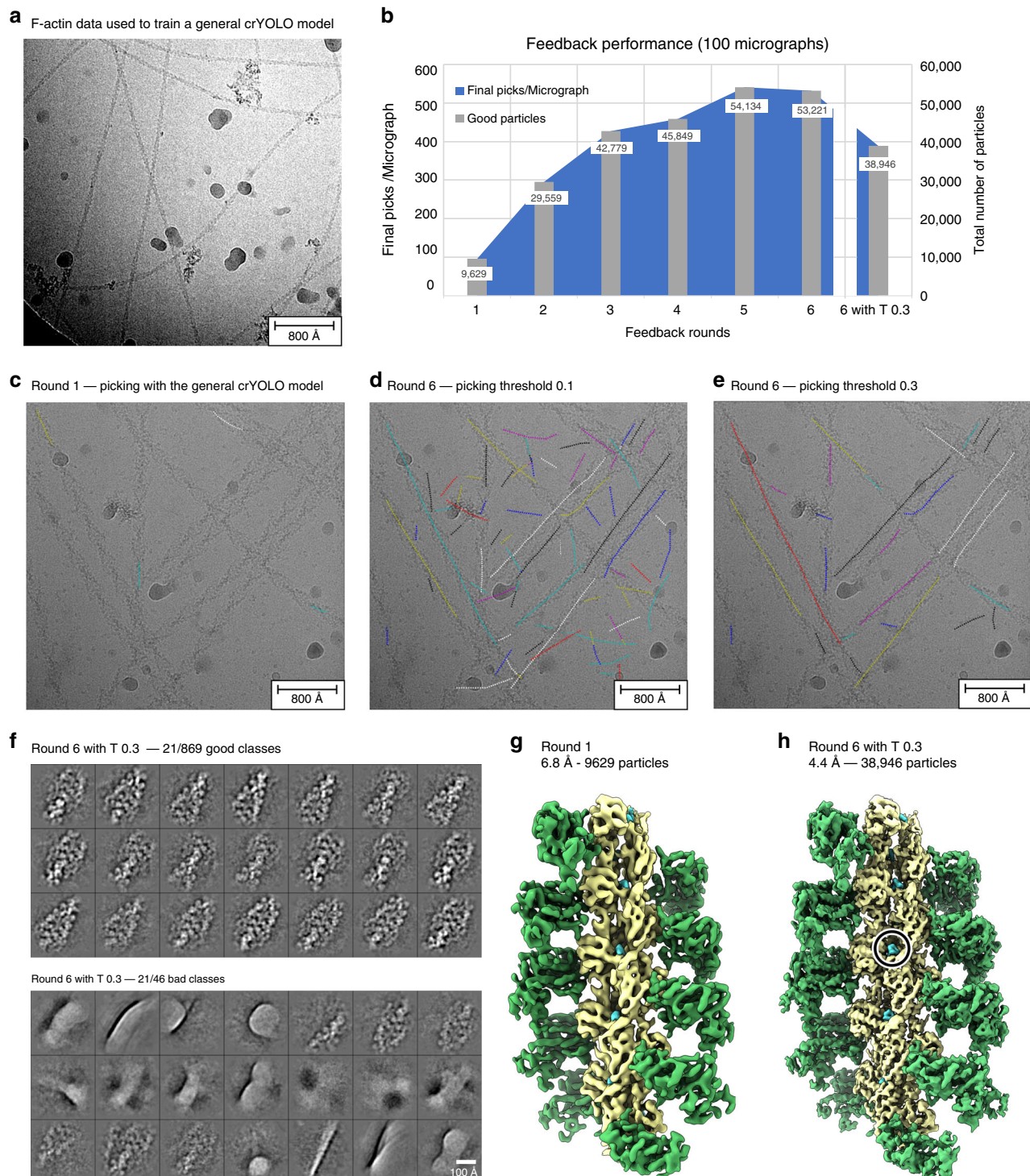

**Fig. 5 Ligand identification within an actomyosin complex. a** Representative micrograph of the F-actin data used to train crYOLO. **b** Progression of the number of "good" particles per micrograph (blue) and in total (gray) when applying the intermediate picking models of the feedback loop to a fixed subset of 100 micrographs. The dipping curve at the end indicates the desired loss of low-quality picks that are excluded when a higher picking threshold (0.3) is used. **c** Representative micrograph of the actomyosin complex highlighting the weak initial picking results when using the crYOLO model trained on F-actin data (see **a**). **d** Particle picking performance on the same micrograph using the final picking model. While filaments are now traced much more effectively, the model also picks unwanted filament crossings and contamination. **e** Increasing the picking threshold from 0.1 to the default value of 0.3 minimizes the amount of false positive picks, while maintaining the desired filament traces. **f** Representative 2D class averages labeled "good" (top) and "bad" (bottom) by Cinderella based on 100 micrographs and using the final model for picking. **g** 3D reconstruction of the actomyosin complex computed from 100 micrographs using the initial picking model. **h** 3D reconstruction computed from the same 100 micrographs using the final optimized picking model. The resolution is sufficient to verify the binding of a ligand (circled).

**Table 5 Summary of the feedback loop statistics for the actomyosin complex.**

| Feedback round | Good classes | Good particles | Picks/Mic | Good picks/Mic | Resolution | Relative good picks |
|---|---|---|---|---|---|---|
| 1 | 211 | 9629 | 143 | 96 | 7.63 | **0.67** |
| 2 | 659 | 29,559 | 358 | 296 | 4.48 | **0.83** |
| 3 | 936 | 42,779 | 552 | 428 | 4.37 | **0.77** |
| 4 | 1016 | 45,849 | 639 | 458 | 4.54 | **0.72** |
| 5 | 1203 | 54,134 | 1098 | 541 | 4.32 | **0.49** |
| 6 | 1174 | 53,221 | 1073 | 532 | 4.72 | **0.50** |
| 6 + T 0.3 | 869 | 38,946 | 515 | 389 | 4.54 | **0.76** |

For every feedback round as well as the final run after optimization of the picking threshold (6 + T x.xx) the number of classes labeled "good" by Cinderella; the number of particles included in these classes; the total number and the number of good particles picked per micrograph; the final resolution of the 3D reconstruction; and the relative amount of good particles (highlighted in bold) are listed for 100 micrographs.

unknown to the pipeline. Moreover, TranSPHIRE now enables experimentalists to produce an early 3D reconstruction with a resolution sufficient to identify bound ligands and determine whether or not their data is likely to yield a high-resolution reconstruction—all within the time frame of hours and while their data is still being collected at the microscope (Fig. 2 and Supplementary Fig. 3). The automated processing greatly simplifies the processing of filamentous samples in general and, most importantly, facilitates the fast determination of multiple structures of one filament decorated with different accessory proteins or bound to ligands.

## Discussion

In this paper, we present the streamlined TranSPHIRE pipeline for automated, feedback-driven processing of cryo-EM data. It fully-automates data transfer, pre-processing, and the creation of a series of early reconstructions based on the most recently processed data (Fig. 1a). At the same time, TranSPHIRE prominently displays all relevant data evaluation metrics, updated in real time (Supplementary Fig. 1), and offers the option to send email notifications when issues are encountered or important milestones—such as the first 2D class averages, or an initial 3D reconstruction—have been reached.

We also introduce the TranSPHIRE feedback loop (Fig. 1b), a machine learning-based method to optimize the internally used particle picking model and adapt our native crYOLO picker to any data set, even while it is still being collected at the microscope. This allows TranSPHIRE to adjust to never before seen data, as well as to avoid any issues that a cryo-EM sample might include, such as unwanted proteins, low contrast, and/or different kinds of contamination. The optimization of the picking model performed by the feedback loop can further be guided by the experimentalist in order to specifically select a subpopulation within the data, such as a distinct conformational or oligomeric state.

We demonstrate these capabilities of TranSPHIRE and its new feedback loop by performing three distinct experiments, each addressing a common issue in cryo-EM: First, we processed the membrane protein TRPC4, after purposefully sabotaging our particle picking to simulate processing a data set that is not only unknown, but initially only barely provides enough useful picks for training. Nevertheless, the TranSPHIRE feedback loop successfully taught crYOLO to identify and pick the sought-after particles without any need for user intervention or expert knowledge input. When the final picking model was then used to automatically compute a full reconstruction, we reached a resolution of 3.6 Å (FSC = 0.143), based on the data extracted from no more than 500 micrographs (Fig. 3).

Second, we processed a sample containing the Tc holotoxin in two different conformational states: The common pre-pore state,

and the significantly rarer pore state that only accounts for about one-fifth of the available particles. In this experiment we injected prior knowledge about the pore state into the pipeline by training Cinderella – our deep learning tool to reject unusable 2D class averages—to only accept class averages of this state. This directed the re-training during the feedback loop and taught crYOLO to focus on the rare pore state particles. As a result, we obtained a picking model that was highly selective for only one conformational state while rejecting not only low-quality 2D class averages, but also high-quality 2D class averages if they displayed the Tc holotoxin in the conformational state that we were not interested in (Fig. 4). This produced a particle stack that was not only populated with an increased number of "good" particles, but also contained less particles overall, as unwanted particles were already rejected during particle picking. The final reconstruction obtained a resolution of 4.2 Å (FSC = 0.143). Such an optimized stack means that any follow-up computations only have to deal with relevant data, allowing for more efficient use of the available computational resources.

Third, we processed a data set of an actomyosin complex to demonstrate how the ability of TranSPHIRE to automatically process cryo-EM data also extends to filamentous proteins. To adjust the pipeline to the processing of filaments, we re-trained both crYOLO and Cinderella in order to teach them about the distinct visual properties of filamentous particles and how to avoid any filament-exclusive pitfalls, such as filament crossings. To specifically showcase the ability of the feedback loop to deal with an initially unknown filament structure, we only taught crYOLO about F-actin, which features a fundamentally different appearance than the actomyosin complex. Cinderella was then only trained with the initial 2D class averages that TranSPHIRE produced during the first iteration of the feedback loop. Despite the initial picking model only knowing about F-actin, Cinderella was able to teach crYOLO about the actomyosin complex and the final reconstruction reached a resolution of 4.4 Å (FSC = 0.143), using the data extracted from merely 100 micrographs (Fig. 5).

In summary, TranSPHIRE offers a fully-automated pipeline that produces highly optimized particle stacks that allow for more effective processing and more efficient use of any available resources, both computational and human. Combined, these features allow experimentalists to make the most of their limited time at the microscope and to identify and address any issues as soon as they surface.

Furthermore, TranSPHIRE produces early reconstructions of proteins, even if initially unknown, thereby enabling experimentalists to assess their data and identify the conformational state of their protein or validate the binding of a ligand, if a ligand-free, high-resolution reference structure is available, while their data is still being collected. Hence, TranSPHIRE allows users to perform automated high-throughput on-the-fly screenings for different buffer conditions or ligands of interest.

## Methods

**Hardware used to run TranSPHIRE**. By default, TranSPHIRE runs on a single machine, which can be combined with a separate workstation or computer cluster to outsource computational power. For the majority of the results presented in this manuscript, a single machine equipped with two Intel(R) Xeon(R) Gold 6128 CPUs (3.40 GHz), featuring 12 CPU cores each (hyperthreading 24); 192 GB of RAM; and three GeForce RTX 1080 Ti GPUs was used. Only computationally more expensive 3D reconstructions, both the initial ab initio reconstruction and the 3D refinements (for details see below), were outsourced to our local computer cluster. There, calculations were performed on two nodes; each equipped with two Intel(R) Xeon(R) Gold 6134 CPUs (3.20 GHz), featuring 32 CPU cores in total and 384 GB of RAM.

**Software integrated into the TranSPHIRE pipeline**. TranSPHIRE is a free of charge, open-source software written in Python 3, which is available online (https://github.com/MPI-Dortmund/transphire). The package can be easily installed via the package manager of python PIP and detailed instructions are available from the TranSPHIRE wiki (https://transphire.readthedocs.io).

Its fully-automated processing pipeline integrates several software packages and is thereby highly flexible and adaptable. An initial integrity check and the consecutive compression of every input stack to a LZW compressed tiff file is performed using IMOD v4.9.8[35]. Currently, TranSPHIRE supports several options for motion correction (Unblur[20] and MotionCor2[19]) and CTF estimation (CTFFIND[21], CTER[22], and GCTF[23]). For all consecutive 2D and 3D processing steps, TranSPHIRE utilizes functions of the SPHIRE[18] package including the deep learning particle picker crYOLO[17], the 2D class selection tool Cinderella[25] and a new GPU accelerated version of the reliable 2D classifier ISAC2[24].

Results presented in this manuscript were generated with TranSPHIRE v1.4.50 and SPHIRE v1.4. Specifically, the pipeline consisted of the following modules: the CUDA 10.2.86 version of MotionCor2 v1.3.0[19]; CTFFIND v4.1.13 for CTF estimation[21]; crYOLO v1.6 for particle picking[17]; SPHIRE sp_window.py for particle extraction[18]; a GPU accelerated version (v1.0) of SPHIRE ISAC2[24] for on-the-fly 2D classification (will be published elsewhere); SPHIRE Cinderella v0.5[25] for 2D class selection; SPHIRE sp_rviper.py[18,26] for ab initio reconstructions and finally SPHIRE sp_meridien.py[18] or sp_meridien_alpha.py for the 3D refinement of single particles or filaments, respectively.

**The automated processing pipeline within TranSPHIRE**. After pre-processing the data i.e. data transfer and compression, motion correction and CTF estimation (also see Supplementary Fig. 2) particles are automatically picked using the deep learning, GPU-accelerated particle picker crYOLO[17]. By using the general model, which was trained on 63 cryo-EM data sets, crYOLO is able to pick previously unseen particles. However, for filamentous data an initial picking model needs to be provided by the user as the general single particle model does not know about filaments. During the feedback rounds a picking threshold of 0.1 is used to facilitate the picking of distinct proteins and features. At the end of each feedback iteration crYOLO is retrained on particles that contributed to classes labeled "good" by Cinderella (see below and Fig. 1b). When crYOLO is trained on a single data set, it quickly reaches a good picking quality even when the training data only contains few micrographs. Hence, increasing the size of the training data, enhances the training time without benefitting the training. Therefore, only particles from 50 randomly selected micrographs and no more than 20,000 particles in total are used for the training. Once the feedback loop is finalized, the picking performance is further optimized by adjusting the picking threshold to an optimal one, as determined by a parameter grid search using crYOLO's internal evaluation procedure. The particle threshold value defines a confidence threshold that each pick made by crYOLO must either meet or exceed in order to be accepted. If this threshold is set to a low value, particles with a low confidence are also accepted. In order to find the optimal threshold, a fixed subset of data is repeatedly picked while varying the threshold from 0.0 to 1.0, using a step size of 0.01. Afterward, the optimal threshold is defined by the highest F2 score[36] of all resulting picks. Processing results generated with the optimized threshold are labeled with iteration "6 + T x.xxx", where six represents the sixth and thus final model used in the feedback loop, and the value x.xxx denotes the optimized picking threshold.

Picked particles are automatically extracted and classified in 2D, resulting in class averages containing 60 to 100 particles per class (standard settings). Classifications are performed by a new GPU-accelerated and updated version of ISAC2, which is based on the original ISAC (Iterative Stable Alignment and Clustering) algorithm[24]. Just like the CPU-bound ISAC2 it delivers high-quality 2D class averages as well as an initial clean-up of the data set, but does not come with the same high computational cost. Hence, GPU ISAC provides the same functionality on a single workstation without the need to outsource 2D classification to a cluster. The GPU ISAC code repository is part of the SPHIRE repository listed above.

As the generation of high-resolution 2D class averages requires a sufficient number of particles covering a range of views, 2D classification is only started once a certain number of particles is accumulated. While this number can be adjusted in the TranSPHIRE GUI, a default value of 20,000 particles per batch has proven to be good (see also Supplementary Fig. 4).

2D class averages are routinely used to assess the overall quality of the data and to select only those particles for 3D refinement that contribute to high-quality 2D class averages. Previously, this selection was done manually, breaking any automated processing pipeline. In order to provide a fully-automated pipeline, TranSPHIRE uses Cinderella[25], a deep learning binary classifier based on a convolutional neural network. When provided with a set of 2D class averages, Cinderella labels each of them as either "good" or "bad." By default, this decision is based on a model that was trained on a large set of class averages from a multitude of different cryo-EM projects. Alternatively, Cinderella can be trained on specific data to select classes according to the needs of the current project. By default, TranSPHIRE runs Cinderella using its general model, based on 3,559 "good" and 2,433 "bad" classes taken from 20 different data sets from both the EMPIAR[37] data base and our in-house efforts. The Cinderella git repository can be found online (https://github.com/MPI-Dortmund/sphire_classes_autoselect).

Once the feedback loop has finished and a set of at least 200 "good" class averages is available (number can be adjusted if desired), a reproducible, ab initio 3D reconstruction is computed from 2D class averages using the SPHIRE method RVIPER[18] (Reproducible Validation of Individual Parameter Reproducibility). The VIPER algorithm combines a genetic algorithm[38] with stochastic hill climbing[39] to produce multiple 3D ab initio structures. These reconstructions are then compared and the most reproducible model is used to seed the consecutive 3D refinement. (See online documentation for RVIPER and VIPER at http://sphire.mpg.de/wiki/doku.php?id=pipeline:viper:sxrviper).

To generate a high-resolution 3D reconstruction a stack of all particles assigned to classes that were labeled "good" by Cinderella is created. The consecutive refinement is performed by the SPHIRE method MERIDIEN[18] providing the initial reconstruction computed in the previous step as reference. The refinement within MERIDIEN proceeds in two phases. The first phase, "EXHAUSTIVE", searches the whole 3D parameter space—three Euler angles for rotation and two dimensions for translation—on a discrete grid. The second phase, "RESTRICTED", searches the parameter space on a discrete grid within the local area closest to the best matching set of parameters found in the previous iteration. To avoid over-fitting, the image dimensions and the grid spacing is adjusted after every iteration, based on the achieved resolution according to the gold standard FSC[40] and stability of the parameters. In order to compensate for the discreteness of the grid and the uncertainty in parameter assignment, particles are weighted by the probability of the parameter set for the backprojection into the 3D reconstruction. (See online documentation of MERIDIEN at http://sphire.mpg.de/wiki/doku.php?id=pipeline:meridien:sxmeridien).

Similar to the prerequisites for 2D classification, a certain number of particles representing different views is required to successfully compute a 3D reconstruction. Thus, TranSPHIRE will not start the 3D refinement before a defined number of particles is accumulated. In our hands, a total of 40,000 particles (default value, can be adjusted) is sufficient to calculate a medium to high-resolution 3D reconstruction in a short time frame. While this reconstruction will likely not reach the highest resolution possible, it still enables a first analysis i.e. identification of a conformational state or the verification if a ligand is bound or not in case a ligand-free high-resolution reference structure is available. Furthermore, it provides quality control throughout the data acquisition, as a new 3D reconstruction is computed for every batch of 40,000 particles. As all 3D refinements start from the same initial reference, refinement projections parameters can additionally be used to directly start with a local refinement of the complete data set, thereby significantly reducing the required running time.

**Evaluation of the feedback performance**. As TranSPHIRE runs in parallel to the data acquisition and data are processed as they come in, the number of movies is increasing during the runtime and results from one feedback iteration to the next are not directly comparable. Thus, the feedback performance was evaluated separately for every data set on a fixed subset of 500 (TRPC4 and Tc holotoxin, Figs. 3 and 4) and 100 (Actomyosin, Fig. 5) micrographs.

For each case, the fixed subset was processed using the intermediate picking models produced during the individual feedback iterations. Specifically, every subset was once picked with the starting model (general model, labeled round 1) and with every picking model generated throughout the five iterations of the feedback loop (rounds 2–6) using a particle threshold of 0.1. In addition, another run was performed with the final picking model using the optimized particle threshold (6 + T X.XX). The consecutive processing in 2D and 3D was performed with AutoSPHIRE sp_auto.py, which is the automatic, batch processing tool within SPHIRE[18] on our local CPU cluster. The processing pipeline and settings used resemble the ones described above, except that CPU ISAC was used instead of the new GPU-accelerated version.

**Automatic processing of the TRPC4 data**. The performance of TranSPHIRE was tested on a subset of 500 micrographs of a high-resolution data set of the transient receptor channel 4 (TRPC4) from zebra fish in LMNG detergent (prepared in analogy to ref. [41]). The data set was automatically collected at a Cs-corrected Titan Krios (FEI Thermo Fisher) microscope equipped with an X-FEG and operated at 300 kV using EPU (FEI Thermo Fisher). Equally dosed frames with a pixel size of 0.85 Å/pixel were collected with a K2 Summit (counting mode, Gatan) direct electron detector in combination with a GIF quantum-energy filter set to a filter width of 20 eV. Each movie contains 50 frames and a total electron dose of 88.5 e/Å$^2$.

Processing in TranSPHIRE was performed as described above with five internal feedback rounds to optimize the crYOLO picking model. Within the pipeline, movies were drift corrected and dose weighted by MotionCor2[19] using five patches with an overlap of 20% and CTFFIND4[21] fitted the CTF between 4 Å and 30 Å with an Cs value of 0.001. The training data for the general model of crYOLO usually contain four data sets of TRP channels. To avoid any favorable picking bias and handle the TRPC4 data as previously unseen, the general model was retrained after removing all TRP channels from the training data. Even then, crYOLO was able to identify most TRPC4 particles through the successful generalization. To simulate a worst-case scenario of a deficient initial picking performance, 90% of the particle picks in the initial feedback round were replaced by random coordinates.

During the feedback rounds the crYOLO picking threshold was set to 0.1 and the anchor size to the estimated particle diameter of 240 pixels. After the final feedback round, the picking threshold value was adjusted to 0.257 based on the crYOLO confidence threshold optimizing procedure described above. After each particle picking step, particles were automatically extracted using SPHIRE sp_window.py with a box size of 288 pixels. The subsequent 2D classification was performed using a GPU accelerated version of the SPHIRE ISAC2 algorithm using standard settings. The feedback loop was run with the default particle batch size of 20,000 (for details see above and Supplementary Fig. 3).

The produced 2D class averages were subjected to an automatic 2D class selection using our deep learning tool Cinderella and a confidence threshold of 0.1. To simulate the processing of a previously unseen protein, Cinderella was trained with its general model training data excluding all channel proteins, thereby ensuring an unbiased selection process. During the feedback rounds crYOLO was trained on the default value of 50 random micrographs that contained particles contributing to classes labeled "good" by Cinderella. 3D reconstructions were computed as described above using no mask and imposing c4 symmetry. Note that albeit our program provides the possibility to compute a 3D mask from the initial model automatically and apply it during the refinement, this option is deactivated by default. Automated masking procedures might eliminate valid regions of the structure that are not well resolved in the initial reconstruction, especially in cases with strong flexibility in the complex. In case a 3D mask is not provided, we strongly recommend to use a mask created from the results of TranSPHIRE for all follow-up experiments, in order to exploit the full potential of 3D refinement. Whereas the workflow can be easily extended, the pipeline for each batch stops by default after the first high-resolution 3D refinement, in order to allow on-the-fly evaluation by the user. The results can be easily converted to RELION[16] after any milestone and vice versa. Correction of higher-order aberrations for example in RELION might further improve the resolution of the final result, when these optical effects are present[42].

The progression of the picking performance throughout the feedback rounds was evaluated on a fixed subset of 500 micrographs as described above (Fig. 3). Note that the picking model of the first iteration is not included in this evaluation, as its performance was initially corrupted by randomizing 90% of the picked particles.

**Automatic processing of the Tc holotoxin data**. To test the capability of TranSPHIRE to target a specific conformation, a subset of 500 micrographs of the ABC holotoxin from *Photorhabdus Luminescens* reconstituted in a lipid nanodisc (EMD-10313)[43] was processed. This data set contains a mixture of conformations, namely the pre-pore and pore state of the holotoxin. The data set was collected at a Cs-corrected Titan Krios (FEI Thermo Fisher) microscope equipped with an X-FEG and operated at 300 kV using EPU (FEI Thermo Fisher). Equally dosed frames with a pixel size of 0.525 Å/pixel were collected with a K2 Summit (super resolution mode, Gatan) direct electron detector in combination with a GIF quantum-energy filter set to a filter width of 20 eV. Each movie contains 40 frames and a total electron dose of 60.8 e/Å$^2$.

Processing in TranSPHIRE was performed as described above with five internal feedback rounds to optimize the crYOLO picking model. Within the pipeline, movies were drift corrected, dose weighted and binned to a pixel size of 1.05 Å/pixel by MotionCor2[19] using three patches without overlap and CTFFIND4[21] fitted the CTF between 4 Å and 30 Å with an Cs value of 0.001. Subsequently, particles were picked using the general model of crYOLO.

During the feedback rounds the crYOLO picking threshold was set to 0.1 and the anchor size to the estimated particle diameter of 205 pixels. After the final feedback round, the picking threshold value was adjusted to 0.194 based on the crYOLO confidence threshold optimizing procedure described above. After each particle picking step, particles were automatically extracted using SPHIRE sp_window.py with a box size of 420 pixels. The subsequent 2D classification was performed using a GPU accelerated version of the SPHIRE ISAC2 algorithm using standard settings. The feedback loop was run with the default particle batch size of 20,000 (for details see above and Supplementary Fig. 3).

The produced 2D class averages were subjected to an automatic 2D class selection using our deep learning tool Cinderella and a confidence threshold of 0.1. To demonstrate the ability of the TranSPHIRE feedback loop to selectively pick particles of one conformational state, Cinderella was trained on pre-existing 2D class averages of the pore state as instances of "good" classes (318) and 2D class averages of the pre-pore state and contamination as instances of "bad" classes (664). During the feedback rounds crYOLO was trained on the default value of 50 random micrographs that contained particles contributing to classes labeled "good"

by Cinderella. 3D reconstructions were computed as described above without applying a mask or symmetry.

The progression of the picking performance throughout the feedback rounds was evaluated on a fixed subset of 500 micrographs as described above (Fig. 4).

**Automatic processing of an actomyosin complex data set**. A subset of 100 micrographs of an actomyosin complex with a bound small molecule ligand was chosen to demonstrate the processing of filamentous samples within TranSPHIRE and its suitability for high-throughput ligand screenings. The data set was collected at a Cs-corrected Titan Krios (FEI Thermo Fisher) microscope equipped with an X-FEG and operated at 300 kV using EPU (FEI Thermo Fisher). Equally dosed frames with a pixel size of 0.55 Å/pixel were collected with a K2 Summit (super resolution mode, Gatan) direct electron detector in combination with a GIF quantum-energy filter set to a filter width of 20 eV. Each movie contains 40 frames and a total electron dose of 81.2 e/Å$^2$.

Processing in TranSPHIRE was performed as described above with five internal feedback rounds to optimize the crYOLO picking model. Within the pipeline, movies were drift corrected, dose weighted and binned to a pixel size of 1.10 Å/pixel by MotionCor2[19] deactivating patch alignment and CTFFIND4[21] fitted the CTF between 5 Å and 30 Å with an Cs value of 0.001.

As the crYOLO general model does not include filamentous data it cannot be readily applied to this data set. Instead a new crYOLO general model specific for actin filaments was trained. The training data consisted of multiple actin data sets collected within our group, but did not include any data of an actomyosin complex or other actin complexes. Considering the significant optical difference of actin and actomyosin filaments (also see Fig. 5), picking with the general actin crYOLO model mimics the processing of a previously unseen filamentous protein.

During the feedback rounds the crYOLO picking threshold was set to 0.1 and the anchor size to the estimated box size of 320 pixels. Furthermore, the filament width was set to 100 pixels and the box distance to 25 pixels (equivalent to one helical rise of 27.5 Å). Only filaments consisting of at least six segments were considered. After the final feedback round, the picking threshold value was adjusted to the crYOLO default value of 0.3, as the threshold optimization procedure of crYOLO does not support filaments. After each particle picking step, particles were automatically extracted using SPHIRE sp_window.py with a box size of 320 pixels and a filament width of 100 pixels. The subsequent 2D classification was performed using a GPU accelerated version of the SPHIRE ISAC2 algorithm asking for 30–50 particles per class. The feedback loop was run with the default particle batch size of 20,000 (for details see above and Supplementary Fig. 3).

The produced 2D class averages were subjected to an automatic 2D class selection using our deep learning tool Cinderella and a confidence threshold of 0.1. As filamentous data differ strongly from the data used to train the general model of Cinderella, a new model was trained based on the 2D class averages produced in the initial feedback round combined with previously selected class averages of actin only data sets. During the feedback rounds crYOLO was trained on the default value of 50 random micrographs that contained particles contributing to classes labeled "good" by Cinderella.

An initial 3D reference was created from a deposited actomyosin atomic model (PDB:5JLH)[44]. The 3D refinement was performed using SPHIRE sp_meridien_alpha.py, an open alpha version of helical processing in SPHIRE, with a particle radius of 144 pixels (~45% of the box size), a filament width of 100 pixels and a helical rise of 27.5 Å. While projection parameters are restrained according to the helical parameters e.g. the shift along the filament axis is restricted to half of the rise no helical symmetry is applied and therefore does not need to be determined beforehand. To avoid artifacts due to the contact of the filament to the edges of the box, a soft 3D mask covering 85% of the filament was applied during the refinement.

The progression of the picking performance throughout the feedback rounds was evaluated on a fixed subset of 100 micrographs as described above (Fig. 5).

## Data availability
The movies processed in this paper are available from the corresponding author upon reasonable request.

## Code availability
TranSPHIRE is open-source and can be downloaded free of charge (https://github.com/MPI-Dortmund/transphire). The TranSPHIRE documentation is available on readthedocs (https://transphire.readthedocs.io).

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

## Acknowledgements

We thank T. Shaik for carefully reading the manuscript and valuable comments, and O. Hofnagel and D. Prumbaum for testing and their continuous feedback. We further thank D. Roderer and D. Vinayagam for providing the Tc holotoxin and TRPC4 data sets, respectively. This work has been funded by the Max Planck Society (S.R.).

## Author contributions

Conceptualization: M.S., T.W., C.G., and S.R.; Software—TranSPHIRE: M.S.; Software—GPU ISAC: F.S.; Software—Cinderella: T.W.; formal analysis: M.S., T.W., S.P.; Writing—Original Draft: F.S.; writing—review and editing: F.S., M.S., S.P., T.W., C.G., S.R.; funding acquisition: S.R.

## Funding

## Competing interests

The authors declare no competing interests.
