## [Peer Review File · Nature Communications]

REVIEWER COMMENTS

Reviewer #1 (Remarks to the Author):

The manuscript presents an automated and data-driven pipeline for cryo-EM on-the-fly data processing, TranSPHIRE. The pipeline streamlines the individual popular tools for motion correction and CTF estimation, as well as some other tools from the SPHIRE package for particle picking, 2D classification/clustering, ab initio modeling and 3D refinement. Notably, the authors introduced a “feedback loop” that uses the results from Cinderella, a classifier that can assess the 2D classification results, to retrain crYOLO, the particle picker. This feedback loop strategy can successfully train a better particle picking model and determine the best determination threshold with minimum user intervention. The pipeline is also optimized well in terms of parallel computing/GPU acceleration, so that it can provide useful feedback and final results in a reasonable amount of time.

We have only have a few minor comments and questions to be addressed in a revised manuscript

1) The authors need to put this work within the context of other pipelines that currently exist such as cryoSPARC Live. The authors do not discuss this nor do they cite a recent publication that utilized deep learning for automated preprocessing - PMID: 32294468.

2) It seems the success of the feedback loop strategy is dependent on the accuracy of Cinderella and data quality. When there is a dataset with many ‘bad’ micrographs and ‘bad’/borderline class averages, how does transSPHIRE handle this? Is it possible to capture this caveat in the pipeline as early as possible? Or does the pipeline have the robustness to overcome this, so that after a few feedbacks, the particle picker and Cinderella will not reinforce the errors made before?

3) Related to #2, the authors should include a description in the text regarding robustness against challenging datasets.

In the case when the user needs/wants to manually pick 2D averages to retrain Cinderella, how much labeling does the user have to do to train a reliable model? In the retraining process, does the user have to tune any parameters to prevent possible overfitting?

Overall, we think this is a well-written manuscript and will be of great interest to the cryoEM community.

Reviewer #2 (Remarks to the Author):

TranSPHIRE reviewing

The manuscript by Stabrin et al. “TranSPHIRE: Automated and feedback-optimized on-the-fly processing for cryo-EM” describes a new full-standing software package for analysis of single-particle cryo-EM datasets performed on-the-fly while collecting the data. While various software or in-house scripting solutions are already available to perform the same task, Stabrin and colleagues introduce a novel and efficient machine learning solution to eliminate completely human intervention or reduce it to the minimum.

I think that the strength of the manuscript resides in this machine learning-based feedback loop that allows the user to pick the best possible dataset and select good 2D classes in a fully automated way. Another important aspect of TransSPHIRE seems the ability to nicely distribute the computational work load even though this aspect is not fully described by the authors. A disadvantage of TransSPHIRE seems to be its strict dependence on the SPHIRE suit, but still the work is solid and definitively very useful and appealing for any electron microscopist. In my view, with some minor inclusions, corrections and explanations this manuscript is amenable for

publication. I describe below more in details parts that needs to be clarified and improved.

1. On the language side, I would avoid having too many adjectives made of multiple words and when doing so I would use hyphens (e.g. close-to-atomic resolution one could use near-atomic instead). But I am not a native speaker so...

2. There are all throughout the text remarks about the need for an automated SPA cryo-EM processing pipeline specifically for drug discovery. The authors seem to be pointing to a use of EM in a way similar to what was done in X-ray crystallography with structural genomics. However, at the resolutions most frequently achieved with fully automated cryo-EM reconstructions, is not yet possible to really have deep insights in drug discovery. The authors should not completely remove references to drug discovery applications of cryo-EM, but they should argue a bit more when they mention it and describe this possible application in a more realistic way.

3. Existing software solutions to perform on-the-fly data processing are all introduced, but some statements are not fully correct. For example (line 52), Focus, Appion and Warp are described as incapable to generate 3D reconstructions, but they are actually often used in combination with cryoSPARC or Relion to achieve that purpose in a way similar to what TransPHIRE does by plugging into SPHIRE capabilities. Another example (line 62), what do the authors mean by saying that Relion and Scipion have accessibility issues for quality metrics?

This manuscript offers very nice new tools (the feedback loop for picking and the 2D automatic quality check) and in my opinion the authors should emphasize these aspects in the comparisons to other software instead of finding unclear pitfalls in the others.

4. In line 74 the authors say "without the need to outsource the computer load to a computer cluster", but then they mention later on in the Methods the need for a local cluster from the ab initio reconstruction on. This is the same that happens with many of the other available solutions (Warp, Scipion) so it is not fair to report it in the end of the Introduction as a unique feature of TransPHIRE.

5. In general, it is not clear overall in the paper how the computational resources are actually distributed, how easy is the installation as well as the definition of the computational load (what tasks are done where). It appears that the load is distributed automatically in the best possible way between the 3 GPUs and the 24 CPUs of the workstation, correct? If so, how is this achieved? It would improve the paper to have a dedicated paragraph that describes these aspects. Figure 2 partially does it but it is unclear to me, if one counts the parallel CPU threads, they sum up to 29, when only 24 are available. Or are threads and CPU 2 different things?

6. The Results thoroughly explain the concept that TransPHIRE makes it possible to "identify and process only those parts of the data that contribute to high-quality results" (line 91). Nice.

7. In line 135 TransPHIRE is described as able to extract a large variety of metadata. It would be important to list them all (maybe on a table?) depending on what software one used for data acquisition. What is extracted from EPU, SerialEM, Legion, Latitude and JADAS? Is any information about beam tilt and beam tilt direction extracted at all? These will be very important in the future.

8. This software clearly benefits from the picker cryOLO and Cinderella both developed by the same group. While cryOLO is extensively used and documented in the community, Cinderella is little known. The code is available and cited in the manuscript, but there is no formal publication about it. This does not make the software less interesting in my view, but I would be interested to know how Cinderella deals with preferred orientations. Can the authors comment on this? Either in the answer only or even in the main text if they deem it reasonable to do? In general, as preferred orientation is one of the main problems when looking at new samples, the authors should comment on it. E.g. should one stop collecting once the 2D classes show only few views? Is there some kind of warning from Cinderella?

9. In general, in the manuscript there are barely any description of pitfalls or problems and how to deal with them. I think showing also possible problems makes a paper stronger. E.g. how does the particle box dimension affect the whole procedure/robustness?

10. Line 199 the right word is "rest" of the data acquisition not "remainder"

11. Line 207 "the aim should be to process as little data as possible". I would also add as little and as good/homogeneous data as possible. It is very nice that the authors highlight the importance of

gathering the best possible sub-dataset and their feedback tool goes on this direction.

12. How easy is to provide an external starting model with rescaling and all? And how easy is to extract particles and their info to feed into other software? How easy is for the user to choose a good starting model? Nice to add a comment on these points in the text.

13. Will the general model of crYOLO include all training for filamentous sample? Or will have to be a different model?

14. The examples reported are convincing, but what happens if we have no prior info about the shape of different conformers but we only know the number of different conformations we expect?

15. The resolution reported for the cases analysed are perfectly good, but the claim that at that resolution (around 4Å) one can screen ligand presence is optimistic. Of course, one can screen for a large enough co-factor presence, but ligands and drugs are usually small molecules.

16. One last thing. Even though all the relevant software info is found in the Methods, it would be nice to have a Table gathering all the links together.

In summary, the manuscript is mostly complete and clear in my view, only some of the suggested addition/comments could make it even better, I hope.

All the best

Marta Carroni

We thank the reviewers for their constructive and positive feedback that helped us to clarify many important aspects and thereby altogether to improve the manuscript. Major modifications of the manuscript are highlighted in yellow. Below we include our detailed response to each point.

Reviewer #1 (Remarks to the Author):

The manuscript presents an automated and data-driven pipeline for cryo-EM on-the-fly data processing, TranSPHIRE. The pipeline streamlines the individual popular tools for motion correction and CTF estimation, as well as some other tools from the SPHIRE package for particle picking, 2D classification/clustering, ab initio modeling and 3D refinement. Notably, the authors introduced a “feedback loop” that uses the results from Cinderella, a classifier that can assess the 2D classification results, to retrain crYOLO, the particle picker. This feedback loop strategy can successfully train a better particle picking model and determine the best determination threshold with minimum user intervention. The pipeline is also optimized well in terms of parallel computing/GPU acceleration, so that it can provide useful feedback and final results in a reasonable amount of time.

We thank the reviewer for acknowledging the quality of our data and optimization strategy and are happy to answer all questions in detail.

We have only have a few minor comments and questions to be addressed in a revised manuscript

1) The authors need to put this work within the context of other pipelines that currently exist such as cryoSPARC Live. The authors do not discuss this nor do they cite a recent publication that utilized deep learning for automated preprocessing - PMID: 32294468.

We have added additional information to the introduction (p.5) to properly put our work in the context of other pipelines. We also mention and cite the respective works regarding deep learning in automated preprocessing. In addition, we avoided to include personal opinions about other software packages as requested by Reviewer 2. We had initially left out cryoSPARC live, because the program seems to be still in a closed beta state and no further information and publication is available. We mention it now, however, without being able to cite a publication.

2) It seems the success of the feedback loop strategy is dependent on the accuracy of Cinderella and data quality. When there is a dataset with many ‘bad’ micrographs and ‘bad’/borderline class averages, how does transSPHIRE handle this? Is it possible to capture this caveat in the pipeline as early as possible? Or does the pipeline have the robustness to overcome this, so that after a few feedbacks, the particle picker and Cinderella will not reinforce the errors made before?

The reviewer assumes correctly that the feedback depends on the accuracy of Cinderella. However, the data quality is of secondary importance, as our 2D classification algorithm ISAC is usually able to deal with it, as we tried to show with our TRPC4 experiment. In this case we simulated an initial “bad” picking performance, as one would encounter in case of a data set of poor quality, and showed that the feedback loop is able to recover to a good picking state. In addition, TranSPHIRE makes it easy to capture possible caveats early on. First, TranSPHIRE offers to send notifications (email or telegram message to your mobile phone) as soon as new 2D class averages are available, thereby enabling the user to check the results of Cinderella immediately. Second, we introduced an interactive Cinderella re-training tool since our initial submission, that allows the user to stop the process, re-train the Cinderella model with all 2D class averages obtained so far, and either restart the specific feedback round, or the whole feedback loop with just one click. To highlight these features we added a paragraph on p. 8/9 to our manuscript:

3) Related to #2, the authors should include a description in the text regarding robustness against challenging datasets.

In the case when the user needs/wants to manually pick 2D averages to retrain Cinderella, how much labeling does the user have to do to train a reliable model? In the retraining process, does the user have to tune any parameters to prevent possible overfitting?

As described in #2, we introduced an interactive built-in Cinderella re-training tool that enables the re-training of Cinderella with minimum effort. The 2D selection results generated with the new Cinderella model are visualized immediately, allowing the user to check the performance and adjust the model when necessary. This feature is

now also described in the revised manuscript (see answer to question #2). Furthermore, we added the following information on the amount of training data needed to create a reliable model to the revised manuscript.

“A reliable model can usually be produced from 40 manually labeled classes, that should be approximately evenly distributed into “good” and “bad” classes.”

The default training parameters have worked for every data set tested so far, and we are of the opinion that further parameter optimization at this point is not necessary. Specifically, hyperparameter optimization is usually necessary for a program in general, but not necessarily for a specific project. Regularization measures like data augmentation and early stopping based on validation data are already in place to prevent overfitting. We addressed the reviewers' remarks in the results to make the information available and clear to the reader:

“For data sets that require re-training of Cinderella, it is usually sufficient to re-train once on the very first set of obtained 2D class averages. However, further re-training can be performed if deemed necessary by the user to adapt to the 2D class averages of higher quality produced in later stages of the feedback loop.”

Reviewer #2 (Remarks to the Author):

TranSPHIRE reviewing

The manuscript by Stabrin et al. "TranSPHIRE: Automated and feedback-optimized on-the-fly processing for cryo-EM" describes a new full-standing software package for analysis of single-particle cryo-EM datasets performed on-the-fly while collecting the data. While various software or in-house scripting solutions are already available to perform the same task, Stabrin and colleagues introduce a novel and efficient machine learning solution to eliminate completely human intervention or reduce it to the minimum.

I think that the strength of the manuscript resides in this machine learning-based feedback loop that allows the user to pick the best possible dataset and select good 2D classes in a fully automated way. Another important aspect of TransSPHIRE seems the ability to nicely distribute the computational work load even though this aspect is not fully described by the authors. A disadvantage of TransSPHIRE seems to be its strict dependence on the SPHIRE suit, but still the work is solid and definitively very useful and appealing for any electron microscopist. In my view, with some minor inclusions, corrections and explanations this manuscript is amenable for publication. I describe below more in details parts that needs to be clarified and improved.

We thank Marta Carroni for highlighting the novelty and strength of our software package and are happy to address any remaining issues.

1. On the language side, I would avoid having too many adjectives made of multiple words and when doing so I would use hyphens (e.g. close-to-atomic resolution one could use near-atomic instead). But I am not a native speaker so...

We agree with the reviewer that the reading flow can be improved by a more concise wording. Thus, we have changed 'close to atomic' to 'near-atomic' and use hyphens for 'high-throughput', 'fully-automated' and 'fully-integrated' throughout the manuscript.

2. There are all throughout the text remarks about the need for an automated SPA cryo-EM processing pipeline specifically for drug discovery. The authors seem to be pointing to a use of EM in a way similar to what was done in X-ray crystallography with structural genomics. However, at the resolutions most frequently achieved with fully automated cryo-EM reconstructions, is not yet possible to really have deep insights in drug discovery. The authors should not completely remove references to drug discovery applications of cryo-EM, but they should argument a bit more when they mention it and describe this possible application in a more realistic way.

We thank the reviewer for pointing out the need for further clarification and added a respective paragraph on p. 2.

Furthermore, we clarified throughout the manuscript that it is only possibly to unequivocally validate the binding of a ligand and identify its binding site, if "a ligand-free high-resolution reference structure is available."

3. Existing software solution to perform on-the-fly data processing are all introduced, but some statements are not fully correct. For example (line 52), Focus, Appion and Warp are described as incapable to generate 3D reconstructions, but they are actually often used in combination with cryoSPARC or Relion to achieve that purpose in a way similar to what TranSPHIRE does by plugging into SPHIRE capabilities. Another example (line 62), what do the authors mean by saying that Relionit and Scipion have accessibility issue for quality metrics? This manuscript offers very nice new tools (the feedback loop for picking and the 2D automatic quality check) and in my opinion the authors should emphasize these aspects in the comparisons to other software instead of finding unclear pitfalls in the others.

We thank the reviewer for the pointing this out and updated the relevant part in the introduction accordingly. First, we removed the paragraph about possible drawbacks of other software and focused on the advantages of TranSPHIRE, rather than on the disadvantages of other software packages. Second, we put TranSPHIRE more into the context of currently available software packages as requested by reviewer 1. Therefore, we added a paragraph on p. 3 to the revised manuscript in order to clarify these points.

4. In line 74 the authors say “without the need to outsource the computer load to a computer cluster”, but then they mention later on in the Methods the need for a local cluster from the ab initio reconstruction on. This is the same that happens with many of the other available solutions (Warp, Scipion) so it is not fair to report it in the end of the Introduction as a unique feature of TranSPHIRE.

In the revised manuscript we now focus more on the strengths of TranSPHIRE with less emphasis on what other packages cannot do. In addition, we removed the sentence “without the need to outsource the computer load to a computer cluster” and added a sentence that TranSPHIRE is able to outsource computation to stronger devices via ssh if needed:

“While TranSPHIRE can run on a single GPU machine, it additionally offers the possibility to outsource the computationally-expensive 3D reconstructions via SSH connection to a separate machine or computer cluster.”

5. In general, it is not clear overall in the paper how the computational resources are actually distributed, how easy is the installation as well as the definition of the computational load (what tasks are done where). It appears that the load is distributed automatically in the best possible way between the 3 GPUs and the 24 CPUs of the workstation, correct? If so, how is this achieved? It would improve the paper to have a dedicated paragraph that describes these aspects. Figure 2 partially does it but it is unclear to me, if one counts the parallel CPU threads, they sum up to 29, when only 24 are available. Or are threads and CPU 2 different things?

We understand the confusion; usually CPUs are not fully occupied all the time. Therefore, CPU use is oversubscribed by a scheduler and the final distribution is handled by the operating system. Therefore, threads and physical CPUs are indeed not identical. We added a respective paragraph to address this issue and explain the distribution of resources in more detail on p. 5.

In addition, we adjusted the figure legend of figure 2 to avoid confusion:

„Due to the internal scheduling of modern operating systems, and because not every TranSPHIRE thread is always working to capacity, the number of available CPUs (12 / 24 hyperthreading) and assigned TranSPHIRE threads (45) is not identical, and does not limit the speed of the computations.“

We furthermore added additional information about how to install TranSPHIRE:

“The package can be easily installed via the package manager of python PIP and detailed instructions are available from the TranSPHIRE WIKI (<https://transphire.readdocs.io/>).”

6. The Results thoroughly explain the concept that TranSHIRE makes it possible to “identify and process only those parts of the data that contribute to high-quality results” (line 91). Nice.

Thank you for the very positive feedback!

7. In line 135 TranSPHIRE is described as able to extract a large variety of metadata. It would be important to list them all (maybe on a table?) depending on what software one used for data acquisition. What is extracted from EPU, SerialEM, Leginon, Latitude and JADAS? Is any information about beam tilt and beam tilt direction extracted at all? These will be very important in the future.

We agree that this information is important and added a list of the extracted meta data depending on the software used for data acquisition (Table 3) to the revised manuscript.

Currently, TranSPHIRE only supports EPU and Latitude S (also stated in the manuscript) as we don't have example data for the other software packages. However, we hope to extend the support to other acquisition software packages soon.

8. This software clearly benefits from the picker cryOLO and Cinderella both developed by the same group. While cryOLO is extensively used and documented in the community, Cinderella is little known. The code is available and cited in the manuscript, but there is no formal publication about it. This does not make the software less interesting in my view, but I would be interested to know how Cinderella deals with preferred orientations.

Can the authors comment on this? Either in the answer only or even in the main text if they deem it reasonable to do? In general, as preferred orientation is one of the main problems when looking at new samples, the authors should comment on it. E.g. should one stop collecting once the 2D classes show only few views? Is there some kind of warning from Cinderella?

We are happy to comment on this question, however, prefer not to discuss Cinderella in greater detail in this manuscript, but rather elaborate it appropriately in a future manuscript, which is in preparation. In general, Cinderella can distinguish “wanted/good” classes from “unwanted/bad” classes based on manually labeled training data. Our general model was trained on a broad spectrum of data sets, and thereby has learned to accept classes which have good contrast and show features typical for proteins and reject blurry classes or those showing common contamination. Consequently, Cinderella has no knowledge about the protein orientation and is also not per-se able to recognize a preferred orientation problem. Therefore, identifying preferred orientation problems still remains a task for the user. But we believe that TranSPHIRE is of great help to do so very early on and take appropriate measures. For example, the user can enhance the picking of rare views by selectively retraining the Cinderella model (a new tool that allows to re-train Cinderella easily is now described in the revised manuscript, also see answer to question 2 of reviewer 1). In some cases, it still might be necessary to stop the data collection and optimize the sample or change the data acquisition strategy e.g. collect tilted images.

9. In general, in the manuscript there are barely any description of pitfalls or problems and how to deal with them. I think showing also possible problems makes a paper stronger. E.g. how does the particle box dimension affect the whole procedure/robustness?

We agree, that discussing possible pitfalls is important and therefore added information about what to do if the performance of the general Cinderella model is not satisfying. Furthermore, we decided to create a wiki page that provides in-depth information about pitfalls and problems and which is now referred to in the revised manuscript. The wiki has the advantage that it can be easily updated and adjusted based on the users' needs and further developments of TranSPHIRE. We added a respective paragraph on p. 8/9.

In the manuscript we do not give any recommendation about the particle box dimensions, as there are no additional requirements imposed by TranSPHIRE compared to the ones every SPA reconstruction has, i.e. that the particle should not occupy more than 2/3 of the box for defocus values below 5 micrometer. Otherwise, one might run into problems with correcting the CTF, as the information is dislocated. If one is unsure about the dimensions of the protein, crYOLO is also able to estimate the protein radius from the picked particles. As we are aware that beginner might struggle with choosing the right box size and/or other relevant parameters, we will link our extensive SPHIRE tutorial, as well as, the wiki page which also covers crYOLO and Cinderella in the above mentioned FAQ section.

10. Line 199 the right word is “rest” of the data acquisition not “remainder”

We checked with native speakers and came to the conclusion that remainder is the correct word in this context.

11. Line 207 “the aim should be to process as little data as possible”. I would also add as little and as good/homogeneous data as possible. It is very nice that the authors highlight the importance of gathering the best possible sub-dataset and their feedback tool goes on this direction.

We agree and added “as homogeneous” to the description.

12. How easy is to provide an external starting model with rescaling and all? And how easy is to extract particles and their info to feed into other software? How easy is for the user to choose a good starting model? Nice to add a comment on these points in the text.

This is indeed a good point. We added a sentence to describe the possibility to scale and clip a starting model to the data at hand:

“In case an external starting model, either map or atomic model, is already available, the SPHIRE and EMAN2 package {Moriya:2017hk, Tang:2007ft} provides straight-forward tools to convert, rescale and clip it to the desired box and pixel size.”

Furthermore, the SPHIRE package contains utilities to convert SPHIRE files into the .star file format, which is readable by most of the other software packages. We added more information about this feature to the manuscript about it:

“To support interoperability with existing packages, all pre-processing steps until particle picking support the file formats used in both SPHIRE and RELION; for later processing steps the SPHIRE package {Moriya:2017hk} provides utilities to easily convert SPHIRE files into RELION {Scheres:2012bs} .star files, which can then be used for further processing in many other cryo-EM software tools.”

13. Will the general model of crYOLO include all training for filamentous sample? Or will have to be a different model?

Unfortunately, the general model for SPA cannot be applied to filamentous samples, as we explained in the result section about filamentous processing:

“While the actual processing is fully automated, some preparation is still needed when using the TranSPHIRE pipeline to process filaments. Specifically, crYOLO needs to be trained to pick filaments, as these look fundamentally different from the single particle complexes known to its default general model.”

To further clarify this point, we added a sentence in the methods section of the revised manuscript:

“However, for filamentous data an initial picking model needs to be provided by the user as the general single particle model does not know about filaments.”

14. The examples reported are convincing, but what happens if we have no prior info about the shape of different conformers but we only know the number of different conformations we expect?

Within TranSPHIRE one can only separate conformations based on 2D classes. Consequently, it is not possible to separate different conformations if they don't look significantly different in 2D. TranSPHIRE would therefore generate a consensus structure of all conformations, which will likely result in an imprecise 3D structure. However, TranSPHIRE also provides a good starting point for a subsequent 3D classification. To alert the reader of this limitation, we added the following sentence to the discussion:

“In case of sample heterogeneity, which cannot be accounted for on a 2D level, it is important to run a 3D sorting procedure after data acquisition.”

15. The resolution reported for the cases analysed are perfectly good, but the claim that at that resolution (around 4Å) one can screen ligand presence is optimistic. Of course, one can screen for a large enough co-factor presence, but ligands and drugs are usually small molecules.

We agree that we should discuss this point in more detail and added a short paragraph about cryo-EM and drug screening to the introduction (p. 2). From our experience, the binding of a small molecule in the range from 400 Da to 800 Da can be verified if a ligand-free high-resolution reference structure is available. However, we agree with the reviewer that the validation without a reference model is difficult > 4 Ångstrom. Therefore, we added the condition “if a ligand-free high-resolution reference structure is present” wherever we discuss the identification of a drug within the revised manuscript.

16. One last thing. Even though all the relevant software info is found in the Methods, it would be nice to have a Table gathering all the links together.

Following this suggestion, we added a new table (Table 2), which lists all available software packages and also indicates if they run on CPUs or GPUs.

In summary, the manuscript is mostly complete and clear in my view, only some of the suggested addition/comments could make it even better, I hope.

All the best

Marta Carroni